# A Wide Field-of-View Light-Field Camera with Adjustable Multiplicity for Practical Applications

**DOI:** 10.3390/s22093455

**Published:** 2022-04-30

**Authors:** Hyun Myung Kim, Young Jin Yoo, Jeong Min Lee, Young Min Song

**Affiliations:** 1School of Electrical Engineering and Computer Science, Gwangju Institute of Science and Technology, 123 Cheomdangwagi-ro, Buk-gu, Gwangju 61005, Korea; gusaud31@gist.ac.kr (H.M.K.); yjyoo89@gist.ac.kr (Y.J.Y.); shcw0328@gist.ac.kr (J.M.L.); 2Artificial Intelligence(I) Graduate School, Gwangju Institute of Science and Technology, 123 Cheomdangwagi-ro, Buk-gu, Gwangju 61005, Korea

**Keywords:** light-field camera, wide field-of-view, micro-lens array, 3D information

## Abstract

The long-fascinated idea of creating 3D images that depict depth information along with color and brightness has been realized with the advent of a light-field camera (LFC). Recently advanced LFCs mainly utilize micro-lens arrays (MLAs) as a key component to acquire rich 3D information, including depth, encoded color, reflectivity, refraction, occlusion, and transparency. The wide field-of-view (FOV) capability of LFCs, which is expected to be of great benefit for extended applications, is obstructed by the fundamental limitations of LFCs. Here, we present a practical strategy for the wide FOV-LFC by adjusting the spacing factor. Multiplicity (*M*) is the inverse magnification of the MLA located between the image plane and the sensor, which was introduced as the overlap ratio between the micro-images. *M* was adopted as a design parameter in several factors of the LFC, and a commercial lens with adjustable FOV was used as the main lens for practicality. The light-field (LF) information was evaluated by considering the pixel resolution and overlapping area in narrow and wide FOV. The *M* was optimized for narrow and wide FOV, respectively, by the trade-off relationship between pixel resolution and geometric resolution. Customized wide FOV-LFCs with different *M* were compared by spatial resolution test and depth information test, and the wide FOV-LFC with optimized *M* provides LF images with high accuracy.

## 1. Introduction

Painters of the Renaissance struggled to depict true colors, light, and the three-dimensional (3D) nature of the objects in their paintings [1,2,3]. To depict a 3D sense, painters such as Vermeer have tried using camera obscura by obtaining the authentic light values [4,5]. Although modern cameras have addressed these issues and can now record 2D projections of 3D objects with accurate color and light value, these 2D projections provide little or no information about the depth of objects in the scene. Painters and photographers have long been fascinated by the idea of realizing 3D images that not only depict color and brightness, but also depict color and brightness and depth information, and Lippmann first realized this aspiration with integral photography [6]. In earnest, light-field (LF) imaging was initiated from the plenoptic function by Adelson [7], and successfully evolved into the first hand-held plenoptic camera by Ng [8], which is called the standard light-field camera (LFC), and the focused LFC by Lumsdaine and Georgeiv [9,10,11,12]. The forefront of such LFCs primarily utilizes a micro-lens array (MLA) as a key component to acquiring rich 3D information on light, including depth, encoded color, specularity, refraction, occlusion, and transparency [13,14,15,16,17]. Post-capture capabilities such as depth estimation, perspective shift, and refocus, well-organized in previous studies [8], have already shown potential beyond conventional cameras. A further capability for a wide field-of-view (FOV) in LFC would be a huge advantage for a wide range of applications including navigation in autonomous vehicles, recognition and tracking, and object segmentation and detection [17,18,19].

The anticipation for wide FOV ability is hampered by rudimentary limitations in terms of sequential capture or camera array approaches in early LFC [20,21]. To overcome static scene capture and bulky/expensive multi-camera arrays, the main limiting factor for building small/low-cost wide FOV LFCs is the hard adaptability of wide-angle lenses to LFCs. As a typical wide-angle lens, fisheye lenses have a fundamentally limited entrance pupil that provides only a very small baseline that prevents the effective capture of LF information. A catadioptric system using curved mirrors is bulky, and for both catadioptric and fisheye, the resolution is essentially limited to that of a single sensor. Monocentric lenses have been employed in wide FOV-LFC systems by rotating imaging systems or arranging multiple image sensors according to Petzval field curvature [22,23] to alleviate these fundamental limitations. Although these LFC systems can successfully capture images with a very wide FOV, bulky relay optic systems or high-priced multiple image sensors deteriorate scalability in realistic applications, requiring portability, cost-effectiveness, and integrative capacity. For practical expandability, conventional LFC systems using a single, flat image sensor have not yet been optimized for wide-angle FOV with single-frame image capture.

Here, we present a facile/practical approach for a wide FOV LFC with single-frame image capture by adjusting the spacing factor. In several factors of LFC, we adopted the multiplicity (*M*) as a design parameter, which is determined by the location of the MLA between the image plane and the sensor plane based on a conventional camera system. For practicality, we employed a commercial lens with adjustable FOV as the main lens. Based on the main lens, the LF information was evaluated considering the pixel resolution and overlapped area according to *M* in each narrow and wide FOV [9,24,25,26]. The layout of MLA in LFC design is divided into two types: Keplerian mode and Galilean mode. In the Keplerian mode, the real image is projected onto the image sensor when the distance of the MLA from the image plane is greater than the focal distance. In the Galilean mode, the virtual image is projected onto the image sensor when the distance of the MLA is closer than the focal distance. Therefore, we employed the Keplerian mode to facilitate *M* adjustment. In Keplerian mode, *M* was optimized in each narrow and wide FOV from the depth accuracy derived from the trade-off relationship with pixel resolution and geometric resolution, calculated according to the baseline by considering several design factors [27,28]. The proposed wide FOV-LFC was implemented with facile customization with MLA and spacer for adjustable *M* in the commercial product. Customized wide FOV-LFCs with different *M* are compared with a spatial resolution test by measuring spatially spread points and a depth information test by acquiring real object images on a checkerboard. The wide FOV-LFC with optimized *M* showed excellent depth estimation with fine sharpness.

## 2. Materials and Methods

A focused light-field camera (LFC) captures a light-field (LF) image by placing the image plane formed from the main lens on the object plane of the micro-lens array (MLA) and projecting it to the image sensor. As shown in Figure 1a, each micro-lens acts like an individual camera and the image objects of adjacent micro-lenses overlap each other on the image plane. By applying stereoscopic to these overlapped images, 3D information can be acquired through various image processes [11,28]. The overlapped area between micro-lens images depends on multiplicity (*M*) = *a/b*, which is the ratio of the distance *a* between the image plane and the MLA and the distance *b* between the MLA and the image sensor. In the case of *M* = 1, the overlapped area does not exist, and the LFC system behaves like a conventional camera, simply acquiring 2D images without light-field information. In the case of *M* being similar to 2, the LF function is possible by overlapping half of the adjacent micro-lens images to cover the entire image transmitted through the main lens. For *M* higher than 2, the overlapped area is more than half of the adjacent micro-lens image and further extends to the micro-lens image beyond the adjacent cell, which can be utilized as a larger baseline in triangulation. As *M* increases, the coverage in a single micro-lens image expands, and consequently, the pixel per degree (PPD) tends to decrease (Figure 1b). Since PPD is inversely proportional to the FOV of the main lens, at a higher FOV, the range of *M* adjustable with PPD is relatively lower than that of lower FOV. Thus, for objects at the same distance, the PPD of FOV 20° is three times greater than the PPD of FOV 60°. Figure 1c shows schematic images of individual lenses with maximum ratios of PPD and overlapping ratio (OR) according to several *M* (1, 2, and 4) in each wide (60°) and narrow (20°) FOV. At 20° FOV, LF image can be acquired by depth estimation with low error with the high overlapping ratio at a relatively high resolution due to a decent maximum ratio of PPD even at relatively high *M*. On the other hand, at 60° FOV, as *M* increases, the maximum ratio of PPD sharply decreases, so the resolution of depth information becomes too low, which causes a large error in-depth estimation. Therefore, a compromise in overlapping ratio with a relatively low *M* is required for an adequate level of PPD at high FOV.

Figure 2a depicts the design of the wide FOV-LFC to adjust *M* with *a* and *b* in Keplerian mode. When the object is at a distance of *a_L_* from the main lens with a focal length of *F_L_*, the image plane is projected at a location with a distance of *b_L_* from the main lens, and this image plane is projected to the image sensor through the MLA to form a projected image as a set of individual micro-images. Therefore, the total distance from the main lens to the image sensor is *B_L_*. The object plane is projected to the image plane through the main lens (wide FOV lens), and this image plane is projected to the image sensor through the MLA to form a projected image as a set of individual micro-images. Disparity estimation is performed with slightly different points of view of each of the micro-images in shared areas between micro-images [27,28]. In more detail, Figure 2b illustrates the projection onto the sensor through the MLA of point images in the image plane for different *a* and *b* ratios (i.e., *M* = 2, 4). *d_ML_* is the diameter of MLA equal to the distance between the centers of adjacent micro-lenses. *P_x1,2_* is the distance between the image point to the principal point of the respective micro-image on the image sensor, and the arrows pointing up and down indicate positive and negative values, respectively. At *M* = 2, the point image is projected by two microlenses, and at *M* = 4, the point image is projected by four microlenses. As *M* increases, disparity estimation is possible, even between micro-images that are farther apart. The baseline distance between micro-lenses is defined as follows:(1)$$d=d_{ML}\times \left(M-1\right)$$

Additionally, *P_x_*, which is the disparity of the point image, is defined as a difference between *P_x_*_1_ and *P_x_*_2_:(2)$$P_{x}=P_{x1}-P_{x2}$$

Since the angles are the same triangles, the following relationship is validated:(3)$$\frac{P_{x}}{b}=\frac{d}{a}$$

The distance of the micro-lens to the object as a camera, that is, the distance *a* between

The image plane and the MLA formed by the main lens, *a,* is described as a function of the distance between the MLA and sensor, *b*, the baseline d, and the disparity *P_x_* as follows:(4)$$a=\frac{d\times b}{P_{x}},$$

The accuracy of a is differentiated by disparity as follows:(5)$$\frac{\partial a}{\partial P_{x}}=\frac{d\times b}{P_{x}^{2}},$$

Substituting Equation (5) for error *a*:(6)$$\partial a=\frac{a^{2}}{d\times b}\partial P_{x},$$

We separate the error into two factors:$$\mathrm{Geometric}\;\mathrm{resolution}=\frac{a^{2}}{d\times b},$$
$$\mathrm{Correspondence}\;\mathrm{error}=\partial P_{x},$$

Geometric resolution is the error in terms of the geometry of the optical setup, and correspondence error is the error that occurs fundamentally in correspondence-matching algorithms [24,29]. Here, we focus on geometric resolution, assuming that the correspondence error is quite small and the matching accuracy is limited to less than one pixel. As shown in Figure 2c, the geometric resolution calculated in the proposed wide FOV-LFC tends to decrease errors as the baseline increases. Baseline is a value obtained by multiplying *M* by the diameter of MLA *d_MLA_*, and geometric resolution is also expressed as a function of *M*. Figure 2d shows the calculated depth accuracy as a function of *M* by dividing the PPD by geometric resolution in each narrow (20°) and wide (60°) FOV. At a FOV of 20°, the depth accuracy is highest at *M* = 6 because a relatively long baseline can be used with a decent level of PPD, even at a higher *M*. At a FOV of 60°, the depth accuracy is highest at *M* = 2 even though the baseline is relatively short because the PPD significantly decreases as *M* increases. For example, in micro-images captured at a FOV of 60°, as *M* increases, sharpness drastically decreases so that the object boundary of the image becomes blurry, resulting in lower depth accuracy.

## 3. Results and Discussion

Figure 3a shows the design scheme of the optical alignment module for the wide FOV-LFC. The main lens mount was the c-mount, usually used in vision systems, and the image plane position was configured in consideration of the flange back (17.5 mm). A commercial image sensor (IMX 178, Sony, Tokyo, Japan) was used, which has a resolution of 3096 × 2080. The distance *b* between the image sensor and the MLA was adjusted by the number of spacers stacked to a thickness of 20 μm per unit to realize the desired *M*. The MLA was firmly fixed while maintaining a constant distance between the image plane and the cover glass using a spring clamp. Figure 3b shows photographs of the optical alignment module. The lens mount and spring clamp were configured to be detachable using clips, and an alignment post was placed to prevent distortion of the optical orientation. Spring clamps were constructed on the four corners to hold the MLA flat at the same level. Figure 3c shows raw micro-images acquired using a wide FOV-LFC module implemented with *M* of 2 (left) and 4 (right), respectively, at a FOV of 60°. The matrix of the microlens array used is hexagonal, so the captured raw micro-image is also arranged in a hexagonal matrix. In *M* = 2, although the overlapping area and baseline are small, each micro-image is seen enough to clearly distinguish the object. On the other hand, in *M* = 4, the overlapping area and the baseline are large, but the clarity of each micro-image is so low that it is difficult to distinguish the object.

Figure 4a shows a schematic of point spread function (PSF) measurement for wide FOV-LFC. The PSF describes the spatial response of an imaging system to a point source [30,31,32]. Multiple spatial point sources are generated with a laser and spatial filter. The spatial filter generates a Gaussian beam as a point source, which removes unwanted multiple aberration peaks and suppresses ring patterns caused by dispersion. With consideration of the objective lens focal length *F*, beam radius input to lens *s*, and laser wavelength *λ*, the pinhole diameter *D* was calculated by the following equation:(7)$$D=\frac{\lambda F}{s}$$

The experimental setup is that the laser has a beam diameter of 1 mm and a wavelength of 630 nm, and the spatial filter consists of a 4× objective lens and a pinhole with a 35 μm diameter (Figure 4b). To measure the PSF change for each different *M*, the LFC was configured with a stage moving perpendicular to the optical axis to measure spatially spread point sources from the on-axis to the off-axis of 3 mm. The PSF measurement images and light intensity distributions in each *M* of 2 and 4 are shown in Figure 4c,d, respectively. In *M* = 2, three spread points were measured uniformly on the on-axis. Although on the off-axis, the intensity tends to decrease with the PSF farther from the optical axis, even at an off-axis of 3 mm, center PFSs were measured with a moderate peak intensity (Figure 4c right). For a wide FOV, the spatial resolution becomes insufficient with a relatively high *M*, so the image quality is difficult to obtain reliably.

In Figure 5, the experimental demonstrations were performed with different M (i.e., M = 2, and 4) to optimize wide FOV-LFC for resolution differences in rendered images, depth maps, and depth resolutions. The distance between the image plane and main lens, *b_L_*, can be calculated by a and known parameters *B_L_* and *b*. In addition, in order to perform depth estimation from the actually extracted disparity, it can be calculated using the parameters of Figure 2a and Equation (4).
(8)$$b_{L}=B_{L}-b-a$$

The object plane distance from the main lens, *a_L_*, is calculated by the thin lens Equation, which is expressed by
(9)$$a_{L}=\left(\frac{1}{f_{L}}-\frac{1}{B_{L}-b-a}\right)$$

Consequently, the depth can be estimated from a disparity that is simultaneously focused on a correspondence point of at least two micro-images. Figure 5a shows the object placement for LF image rendering and depth map extraction on a checkerboard floor with dimensions of 10 cm × 10 cm per unit. Figure 5b shows raw data of micro-lens images with *M* = 2 at a FOV of 60°. In the case of *M* = 2, the patch size is large, and most micro-images are used for rendering (Figure 5c), whereas at *M* = 4, the image patch size is so small that the number of pixels used to render the image is significantly reduced compared to the number of micro-images (Figure 5d). To evaluate the disparity, we used the open-source Plenoptic Toolbox 2.0 modified to fit our LFC and MLA, and the corresponding matching method used the sum of absolute differences (SAD) algorithm [33]. Although this study focused on the hardware aspect and did not consider distortion, a lot of research was conducted on radial distortion correction for MLA as well as the main lens in light-field cameras, so it was possible to improve the performance of depth extraction [34,35]. In depth maps, in *M* = 2, depth information with accuracy high enough to distinguish objects, including the checkerboard pattern on the floor from near to far was extracted (Figure 5e). On the other hand, in *M* = 4, the checkerboard pattern of the floor was only visible in close range, and the objects and the nearby floor were blurred, making it difficult to distinguish depth information (Figure 5f). For depth resolution analysis, the checkerboard was located 30 cm away from the wide-FOV LFC, and the disparity was measured while moving 5 cm to 60 cm in the optical axis direction. Figure 5g is the calculation result of disparity according to distance. Typically, disparity tends to decrease as the distance increases, and the amount of change in disparity also tends to decrease as the measurement distance increases. This implies that the closer the measurement distance, the higher the depth resolution, and the greater the measurement distance, the lower the depth resolution. Even in the disparity plot according to distance, since the parallax trend from near to far is maintained at *M* = 2, depth information can be extracted with high accuracy (Figure 5g,h). However, in *M* = 4, the linear correlation of disparity with the distance is acceptable only in the close range, and as the distance increases, the disparity becomes saturated, making it difficult to obtain accurate depth information (Figure 5i). The subpixel unit used in disparity estimation was set to 0.2 pixels, and when the object was 30 cm to 35 cm, the depth step started at nine and gradually decreased at *M* = 2 (Figure 5j). On the other hand, when *M* was 4, the depth step decreased from 1.5 to 1 in the 30 cm to 35 cm section, making it difficult to estimate the depth.

## 4. Conclusions

In summary, we proposed a wide FOV-LFC by introducing an adjustable *M* as a spacing factor for practical application. Based on a main commercial lens alterable with a wide FOV, LF imaging was realized by optimizing *M* as a design parameter for narrow and wide FOV. A practical design rule was established in the optimization process to evaluate depth accuracy by appropriately compromising geometric resolution and pixel resolution. Based on these realistic considerations, the wide FOV-LFC was customized with a facile approach for adjustable *M* with spacer stacking into a conventional LFC system without significant structural modifications to overcome fundamental limitations. Compared to the previously reported sequential or multiple capture-based systems, the optimized wide-FOV LFC with adjustable M was successfully implemented as a single image-capture method with a high FOV per frame (Table 1). Customized wide FOV-LFC for optimized *M* opens up possibilities for practical applications of wide FOV-LF imaging through PSF measurement and depth mapping of real objects on the checkerboard. Moreover, this compromise strategy further expands conventional LFC systems adapting diverse optical components without huge module changes. We further envision that our facile approach holds great potential for various practical applications requiring 3D information acquisition, such as autonomous vehicles, unmanned aerial vehicles, and autonomous underwater vehicles, without laborious redesign of LFC systems.

## Figures and Tables

**Figure 1 sensors-22-03455-f001:**
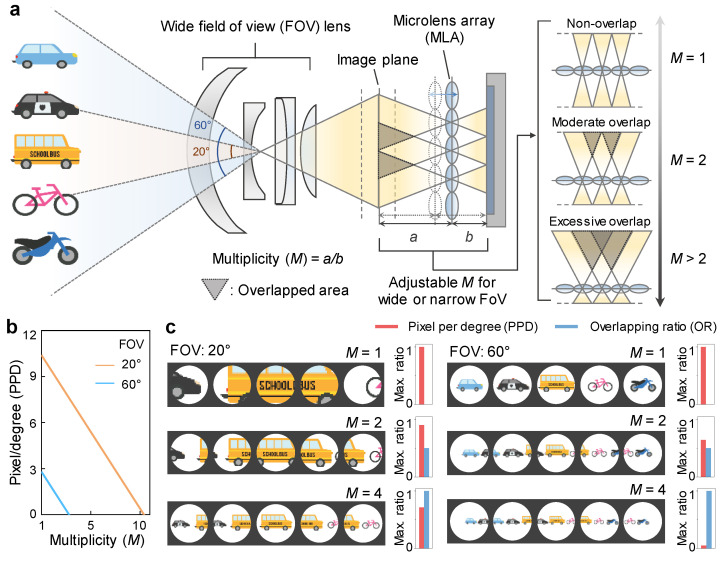
(**a**) Schematic of the proposed light-field camera (LFC) for wide field-of-view (FOV) with several objects. The overlapped area is optimized with adjustable multiplicity (*M*) for wide or narrow FOV. (**b**) Linear plot of pixel per degree (PPD) versus *M* for wide (60°) and narrow (20°) FOV. (**c**) Schematic images of individual lenses with maximum ratios of PPD and overlapping ratio (OR) according to several *M* (1, 2, and 4) in each wide (60°) and narrow (20°) FOV.

**Figure 2 sensors-22-03455-f002:**
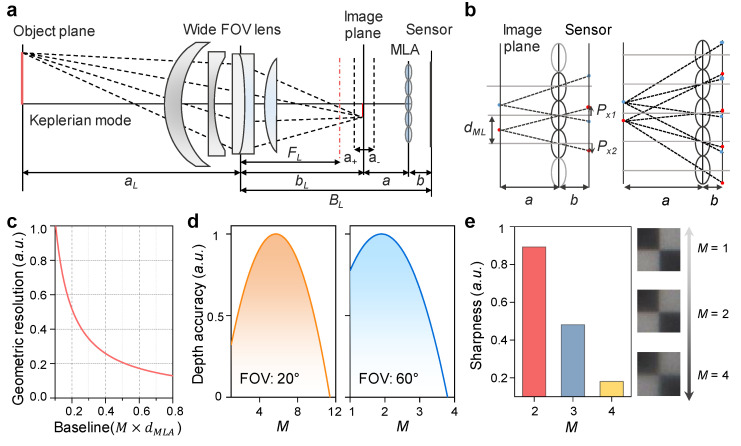
(**a**) Schematic of wide FOV-LFC optical structure with several design parameters in Keplerian mode. (**b**) Schematic of ray propagation path between the image sensor and image plane with different *M*. (**c**) Calculated disparity error according to as a function of baseline from micro-lens projection image. (**d**) Calculated depth accuracy considering PPD and baseline as *M* variation at each narrow (20°) and wide (60°) FOV, respectively. (**e**) Sharpness from captured images according to different *M* (2, 3, and 4).

**Figure 3 sensors-22-03455-f003:**
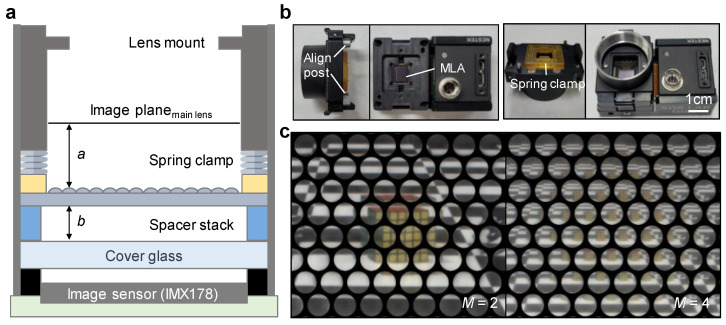
(**a**) Schematic of optical alignment module for wide FOV-LFC. (**b**) Photographs of the spacer-adjustable optical alignment module for wide FOV-LFC. (**c**) Micro-lens images captured by wide FOV-LFC with multiplicity *M* = 2 (left), and *M* = 4 (right).

**Figure 4 sensors-22-03455-f004:**
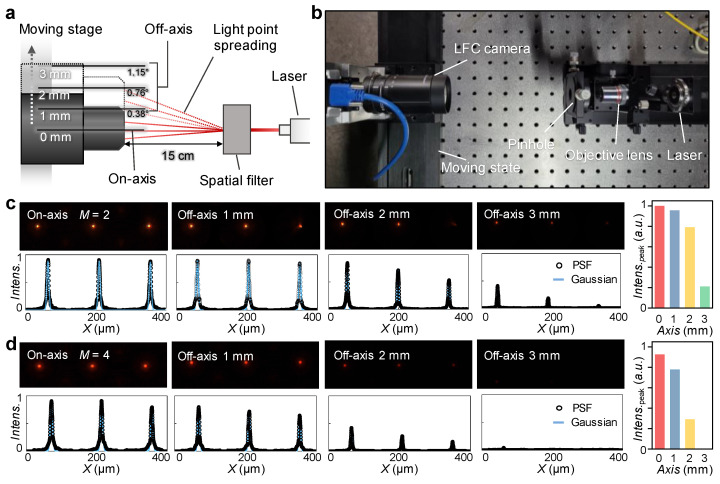
(**a**) Schematic of point spread function (PSF) measurement with axis moving stage. (**b**) Photograph of an experimental set up with a laser beam diameter of 1 mm, a pinhole diameter of 35 µm, 4× objective lens. (**c**,**d**) Captured PSF images (top) and illuminance intensity distribution of PSF (bottom), peak intensities (right) according to axis moving with 0–3 mm at (**c**) *M* = 2, and (**d**) *M* = 4.

**Figure 5 sensors-22-03455-f005:**
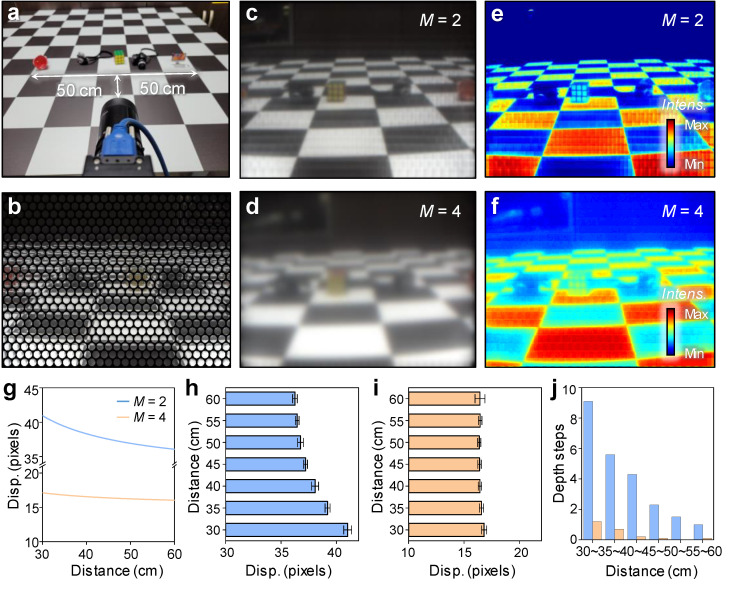
(**a**) Photograph of object-measurement set up of wide FOV-LFC. (**b**) Law data image captured from wide FOV-LFC (**c**,**d**) Preprocessed images from captured object image. at (**b**) *M* = 2 and (**c**) *M* = 4. (**e**,**f**) Disparity maps extracted from captured object images at (**e**) *M* = 2 and (**f**) *M* = 4. (**g**) Calibration result of disparity plots with different distances at *M* = 2 is shown by the blue line, and *M* = 4 the apricot line. (**h**,**i**) Disparity plots with different distances extracted from captured object images at (**h**) *M* = 2 and (**i**) *M* = 4. (**j**) Depth step plot by distance section of object. *M* = 2 is blue bar, and *M* = 4 is apricot bar.

**Table 1 sensors-22-03455-t001:** Various state-of-the-art light-field cameras for wide field-of-view.

Technique	Lens	Additional Components	Num. of Image Sensors	Image Acquisition (Frames)	Field-of-View (FOV per Frame)	Ref
Panoramic single aperture	Monocentric lens	Relay optics, Horizontal rotating stage	1	Sequential capture (11)	138° (24°)	[23]
Panoramic monocentric	Monocentric lens	Multiple consolidators, Multiple fiber bundles	5	Multiple capture (5)	140° (32°)	[20]
Axial light field	Conventional lens	Spherical mirror, Rotating stage	1	Sequential capture (25)	140° (32°)	[36]
Gaussian image blending	Multiple conventional lenses	Hemispherical support, FPGA board	15	Multiple capture (15)	360° (36°)	[37]
Omni-directional light-field imaging	Multiple conventional lenses	Hemispherical support, FPGA board	44	Multiple capture (44)	360° (53°)	[38]
Adjustable multiplicity	Conventional lens	Adjustable spacer	1	Single capture	60°	This work

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
