# Peer review of "A Wide Field-of-View Light-Field Camera with Adjustable Multiplicity for Practical Applications"

_sensors, 2022, doi:10.3390/s22093455_

Round 1

Reviewer 1 Report

Reviewers’ comments on ‘A Wide Field of View Light Field Camera with Adjustable Multiplicity for Practical Applications’ by Kim et al.

The authors have manufactured a plenoptic camera using a standard camera lens and CCD detector.  A microlens array was added between the main lens focus and the CCD detector.  The microlens provides images for multiple varying ray angles incoming to the main lens.  From the angular variation of rays, it is possible to obtain depth information and to achieve focussed images for all depths.  The authors have measured the point spread function of rays at different angles using a laser and scattering surface (labelled ‘spatial filter’ in Figure 4a).  Results for magnifications of 2 and 4 (labelled M) for the imaging from the microlens array are presented.  As the magnification M increases, the different microlens images overlap.  They are also present some data (figure 5) evidencing the depth resolution.

It is not clear to me if there is sufficient novelty in the plenoptic camera and the measurements to warrant publication.  Assuming the work is sufficiently novel I recommend that the authors significantly improve the clarity of the presentation of the paper.  My suggestions are:

  1. In line 71, the authors refer to ‘Keplerian mode’ which must mean that the microlenses are positive lenses. This should be clarified.
  2. The role and behaviour of the ‘spatial filter’ in Figure 4a should be clarified. It is possible that a 35 micron pinhole is insufficiently small and that the images of Figure 4c and 4d are  dominated by the 35 micron diameter of the light scattering ‘point’?  
  3. The angles of light in the point spread function measurement are delineated as distance along the detector and the distance from the ‘spatial filter’ to the detector is not given. I suggest that the angles of the light rays are directly specified in figure 4c and 4d.
  4. Can the authors clarify the ‘disparity maps’ of figure 5. The plots show evidence of depth perception by the movement of an image with distance on the detector, but the significance of the results is not clear.
  5. In the abstract and conclusion the parameter M is introduced without explanation. This parameter should be defined in both cases.  The abstract in particular should be understandable without reference to the main body of the paper.
  6. The authors have not presented samples of the depth resolution capabilities of the camera. I would have expected see some 3D resolution capability evidenced using a 3D object.

Author Response

We appreciate the reviewer for the critical comments, and we have revised the entire text and figures to express the novelty of our work better. Further, we configured a comparison table with other works to clarify the novelty of our work (Table 1). Briefly, the newly developed FOV-LFC in our work has the advantages of simple and effective customization based on commercial products for wide FOV-LFC implementation compared to previously reported techniques. We highlighted these advantages and the potential of our work, and the detailed revisions and additional data are included in the following comments.

Reviewer 2 Report

Manuscript number:  sensors-1668205

Title:  A Wide Field of View Light Field Camera with Adjustable Multiplicity for Practical Applications

Authors: Hyun Myung Kim, Young Jin Yoo, Jeong Min Lee and Young Min Song

The wide field of view light field Camera with adjustable multiplicity is an interesting Job. I agree with the concepts explained in the paper. However, some matters are no described to establish the powerful of the proposed study. In this way, comments and results should be included in the manuscript.

1.- Firstly, the camera model via perspective projection is not described in the manuscript.  Comments about this matter should be included.

2.- The micro lenses are arranged in one dimension to describe the image formation. However, the wide FOV lens is two dimensional, therefore, a micro lens matrix should be arranged. Comments about this matter should be included.

3.- The disparity is mentioned, but the procedure to compute the disparity is not described.  Also, pattern correspondence is not determined.  Comments about this matter should be included.

4.-The distortion is not included to determine the object position in the image plane. Comments about this matter should be included.

5.-The method to optimize the multiplicity M is not described in the manuscript.  Comments about this matter should be included.   

6.-How is computed the image quality at 20° and at 60°?. Comments about this matter should be included.

Author Response

We sincerely appreciate the reviewer for valuable and critical comments that were helpful to improve the quality of our manuscript significantly. We modified our manuscript according to the reviewer’s comments.

Reviewer 3 Report

The wide field of view (FOV) is obstructed by the fundamental limitations of LFCs.   the authors present an effective method for the wide FOV-LFC by introducing an adjustable spacing factor(M) which was optimized for narrow and wide FOV, respectively. The efficiency of the reported method was demonstrated with numerical simulations and experimental measurements . In my opinion, the calculations are correct and the results are convinced. The obtained results may find applications in  wide FOV-LF imaging. I recommend a publication.

Author Response

We appreciate the reviewer's deep insight. we are very pleased that the reviewer understood exactly what we intended. Thank you for your positive comments.

Round 2

Reviewer 1 Report

The authors have revised the paper 'A wide field of view ...' by Kim et al.  The authors have followed the comments of my first review and added text to the manuscript to further explain points raised in my first review.  In particular, the authors have endeavored to quantify the novelty of their plenoptic camera.  I recommend publication.